# Microbial Pigments: Major Groups and Industrial Applications

**DOI:** 10.3390/microorganisms11122920

**Published:** 2023-12-04

**Authors:** João Vitor de Oliveira Barreto, Livia Marques Casanova, Athayde Neves Junior, Maria Cristina Pinheiro Pereira Reis-Mansur, Alane Beatriz Vermelho

**Affiliations:** Bioinovar Laboratory, Institute of Microbiology Paulo de Goes, Federal University of Rio de Janeiro, Rio de Janeiro 21941-902, Brazil; joaobarreto@micro.ufrj.br (J.V.d.O.B.); livia.casanova@yahoo.com.br (L.M.C.); ajunior@micro.ufrj.br (A.N.J.); mariacristinareis@micro.ufrj.br (M.C.P.P.R.-M.)

**Keywords:** microbial pigments, biocolors, organic pigments, biodegradable pigments, biotechnology

## Abstract

Microbial pigments have many structures and functions with excellent characteristics, such as being biodegradable, non-toxic, and ecologically friendly, constituting an important source of pigments. Industrial production presents a bottleneck in production cost that restricts large-scale commercialization. However, microbial pigments are progressively gaining popularity because of their health advantages. The development of metabolic engineering and cost reduction of the bioprocess using industry by-products opened possibilities for cost and quality improvements in all production phases. We are thus addressing several points related to microbial pigments, including the major classes and structures found, the advantages of use, the biotechnological applications in different industrial sectors, their characteristics, and their impacts on the environment and society.

## 1. Introduction

Colors are one of the first parameters, if not the first, captured and evaluated by human sight right after seeing any object, i.e., food, flowers and plants, clothes, animals, and other human beings. Due to this, the most varied industries invest in research and development to create new colors and sources and then conceive new market trends leading to innovation [1,2].

Synthetic pigments have participated significantly in the pigment industry for decades despite their negative impact on the environment and people’s health. Although they dominate the market, microbial pigments, produced as primary or secondary metabolites, are an alternative to synthetic pigments that are environment-friendly and non-toxic. In this context, microbial pigments possess excellent nutritional and therapeutic properties, and the industry is progressively recognizing these properties through the findings published in the literature. For instance, microalgae pigments reached significant commercialization in the food and supplements industry [3]. Microbial pigments are under constant development, and pharmaceutical and textile companies are also commercializing these pigments with a broad possibility of expansion for the cosmetic sector and paint, among others [4]. In addition, advances in organic chemistry and metabolic engineering linked to synthetic biology and omics sciences have enabled large-scale production [5].

## 2. Pigments and Dyes

Pigments or dyes are chemical substances that selectively absorb light and reflect it at a 400–800 nm wavelength, making a specific color visible to the naked eye. Physiologically, these resulting colors integrate first human perceptions, part of a primary system in visual information processing. This process occurs in the photoreceptors and ganglion cells of the human retina, which differentiate such colors in several micro divisions from different shades, nuances, clarities, and saturations. Humans attribute to colors not only everyday perceptions but also a representation of feelings, organization, categorization of objects, and construction of personal styles, among others [6,7]. The functions performed by pigments in living organisms range from beautification as a sensory attractant for pollination to internal functions such as protection against oxidative stress caused by ultraviolet (UV) radiation. These bioactive properties arouse the interest of industries once they are natural compounds with numerous benefits to human health, such as antioxidant, anti-inflammatory, and antimicrobial properties and, sometimes, being associated with the control and avoidance of some chronic diseases, such as types of cancers and cardiovascular problems [8,9]. Differences occur between dyes and pigments, although they have similar applications in providing colors to objects. Dyes are water-soluble compounds with a molecular level of dispersion that confers them brighter and more vivid colors but low stability to light. Meanwhile, pigments are hardly soluble in water by nature, reaching a level of dispersion in particles; they are not affected by the vehicle of incorporation, maintaining the same absorption and mirroring of light [10,11].

## 3. Synthetic and Natural Pigments

Pigments can be synthetic or natural (Figure 1). Synthetic pigments are artificially created. They present various colors and are commonly used in industries, including paints, coatings, plastics, and ceramics. Some examples are cadmium red, iron oxide yellow, black, and bismuth vanadate. Some synthetic inorganic pigments like titanium dioxide and chromium oxide are mineral-based. These pigments are known for their stability and durability [10,11]. Synthetic pigment presents a stable and well-known synthesis process and low-cost production, thus providing an economic advantage. For example, a synthetic orange/yellow pigment, Tartrazine, costs 100 gm 700–800 USD, while microbial carotenoid costs 100 gm 1000 USD. The cost depends on the pigment, the extraction, and all bioprocess conditions, including those upstream and downstream ([3,12]).

With low or no biodegradability, synthetic pigments are contributing to the planet’s unsustainable production chains. In addition, there are reports of cases associated with consuming synthetic pigments with carcinogenic properties and allergic reactions [13,14]. Several synthetic pigments that offer color for food are toxic in the food industry. Hyperactivity in children, allergenicity, toxicological, and carcinogenicity problems have led to the exclusion of many synthetic food pigments [3].

Other classes are synthetic organic pigments, which form a large group of differentiated compounds due to their chemical and physical properties, such as chromogen structure and solubility. We can cite the chromogens indigoid, acridine, and anthraquinones [15]. Synthetic organic pigments are underestimated as environmental contaminants; currently, they are considered micropollutants due to their low concentration (ng/L to µg/L) in aquatic ecosystems [15].

Kant reported that the textile industry is very polluting. The toxicity of synthetic organic and inorganic pigments has become a problem for all life forms [16]. The presence of sulfur, naphthol, chromium compounds, heavy metals (copper, arsenic, lead, cadmium, mercury, nickel, and cobalt), and other chemicals make the textile effluent highly toxic. Other harmful chemicals in the water may be formaldehyde-based pigment-fixing agents, chlorinated stain removers, hydrocarbon-based softeners, and non-biodegradable dyeing chemicals. These organic materials react with many disinfectants, especially chlorine, and often form carcinogenic by-products. The same heavy metal contamination was detected after the use of synthetic pigments in the textile industry in Indonesia [17].

These industrial pollutants were hazardous and carcinogenic for consumers as well. It was not until recently that the effects of synthetic pigments started to surface, and industries were forced to look in a new direction [4]. Eco-friendly products have been valued by the market and the public, who prefer more natural-based options, preferably with philosophies aligned with the planet’s sustainability. This factor even warns traditional brands to support innovation values, bringing sustainable development to attract and retain more consumers [13].

Another type is the natural inorganic pigments such as umber from clay minerals. It has been used in art and as a coloring agent for centuries. Other pigments are azurite and red earth pigment [18]. Pigments from minerals are inorganic and usually extracted from rocks, soils, and ores [19,20]. However, as cited in this review, some synthetic inorganic pigments are mineral-based and chemically synthesized. For example, iron oxide can be classified as a natural inorganic pigment if extracted from rocks or synthetic if chemically synthesized [21].

Natural pigments from mineral sources may contain heavy metal contamination, which could be a problem when used for industrial purposes. Accordingly, numerous regulations worldwide aim to control the presence of lead, arsenic, and other heavy metals brought directly from pigments in some products such as cosmetics [13,22]. The use of mineral pigments promotes ecological risks. In the case of irrigation water, it contaminates agriculture and food for human consumption. If not adequately routed, waste and disposal of this production process can enter a trophic chain and bioaccumulate in fish and aquatic bodies [23].

In the dyes and pigments industry, around 15% of its world production is estimated to be wasted along the production chain up to the application, representing around 128 tons per day of environmental impact. Precisely inorganic pigments, a large part of pigments, become environmental pollutants since they are poorly biodegradable and are bioaccumulated [24,25]. Even at low concentrations, dyes and pigments discarded in rivers and seawater modify the tone of the water, reducing light penetration in the aqueous phase and thus negatively impacting a whole chain of marine life that depends on photosynthetic organisms. The problem becomes evident when additives from the textile industries are considered, using light-stable but nonbiodegradable dyes. In addition, heavy metals can still be discarded, even in underreported amounts, reaching aquatic life and soil and then incorporated into agriculture and cultivated foods, becoming part of the food chain.

Furthermore, what is debatable is how some of these pigments are destined for the cosmetics industry, which should prioritize the use of products that use renewable sources of substrates with a firm policy regarding pollutant disposal during production [24,26,27,28]. Natural organic pigments are growing more expressively as a source of natural pigments. They will be the focus of the next section.

Plant-based pigments are rapidly denatured in the presence of a change in pH, harming reproducibility [29]. In addition, plant pigments require large monoculture areas, pest control, and rainfall reliance, among other circumstantial parameters that affect the quality and the percentage of crop loss [1,30,31,32]. Therefore, this possibility, as much as it is natural, organic, and biodegradable, still causes some degree of impact on the environment due to the need for large irrigation volumes of water and, sometimes, the use of pesticides to help in pest control [13,23].

There are also pigments extracted from animals, mainly from insects and mollusks, which were widely used in ancient times. However, their use has declined over the years due to the difficulty of production and animal exploitation [33]. Insects are a source of several pigments, such as anthraquinones, aphids, pterins, tetrapyrroles, ommochromes, melanins, and papiliochromes [34]. However, obtaining these pigments requires costly insect cultivation and purification [35]. In addition, allergic problems have been reported [36]. Table 1 compares the advantages and disadvantages of three primary pigment sources.

## 4. Microbial Pigments: Major Chemical Groups and Functions

An increasing search for sustainable and non-pollutant bioproducts is a reality in the current planet scenario. Microorganisms can be an excellent source of bioproducts with applications in various industrial sectors. One of these possibilities lies in producing microbial pigments. The use of microorganisms presents numerous advantages. They can grow in bioreactors under physical-chemical-controlled conditions and use industrial residues and by-products as carbon and nitrogen sources for fermentation, reducing production costs [40]. In addition, this procedure lowers the industrial rejects, which will be biotransformed by microbial metabolism [41,42]. The production does not compete for arable land and is not influenced by climate change [43]. Through synthetic biology, they can be modified genetically for optimization for production, including expressing genes in model microorganisms such as *E. coli* [2,4,44,45,46,47]. Microbial pigments are an essential alternative to traditional synthetic pigments in the environment.

Bacteria (Bacteria and Archaea domains), fungi, and microalgae produce microbial pigments. They constitute a broad field with potential biotechnological applications for coloring in the food, cosmetic, textile, and pharmaceutical industries. The microbial pigments are involved in several functions, including survival and adaption to their environment, protection against ultraviolet radiation and reactive oxygen species, antibacterial and fungicidal effects to ensure the territoriality of the environment and nutrients as well preventing the installation of other microbial species in their habitat. Some pigments can also assist in photosynthetic functions in energy production within cells [48,49,50].

A wide variety of structures and functions with different properties are found in microbial pigments, as grouped in a literature review [51]. In a recent review, Agarwal et al. [4] showed the carotenoid-containing biomass from many different microorganisms, such as, for example, *Haematococcus* sp. and *Chlorella* sp. microalgae produced astaxanthin and lutein, respectively. In the present scenario, microbe-derived pigments can take over the global market and cut off synthetic and plant-derived pigments.

### 4.1. Isoprenoids Pigments

Carotenoids, a group of isoprenoid-derived substances, represent nature’s most widely distributed pigments, with yellow, orange, red, and purple colors. They are biosynthesized by archaea, eubacteria, algae, fungi, and plants [52,53,54]. Bright colors in some animals, such as crustaceans, birds, and insects, can be due to carotenoids; however, those are obtained through diet or symbiotic/pathogenic microorganisms [7,53].

Most carotenoids are derived from the precursor phytoene (C_40_). However, some species of bacteria can produce C_50_ and C_30_ carotenoids from different precursors [7]. Their general structure comprises a polyene chain with nine or more conjugated double bonds and an end group on both sides [7,54]. These chains absorb light between 450–570 nm, corresponding to chlorophyll’s absorption gap. Thus, carotenoids can be accessory pigments in photosynthesis [7]. Carotenoid structures are classified into two broad groups: carotenes, non-oxygenated substances, and xanthophylls, corresponding to oxygenated molecules [53,54,55]. Examples of both groups are shown in Figure 2.

The biosynthetic pathway for carotenoids starts with condensing two geranyl pyrophosphate (GGPP) molecules catalyzed by the enzyme phytoene synthase to produce phytoene. This precursor undergoes a series of desaturations and isomerization to form lycopene, a red-colored pigment, which can be later cyclized under the action of cyclases to form molecules such as α-, β-, and γ-carotenes. Hydroxylases, ketolases, or other enzymes may be oxygenated by carotene molecules to originate xanthophylls [55,56]. GGPP can be derived from C_5_ precursors (dimethylallyl diphosphate and isopentenyl diphosphate) from the mevalonic acid (MVA) pathway or the 2-*C*-methyl-D-erythritol-4-phosphate (MEP) pathway, depending on the organism. Archaea, fungi, and most bacteria utilize the MVA pathway, while photosynthetic organisms apply the MEP pathway [55,56]. Figure 2 summarizes the biosynthetic pathway for carotenoids.

Many microorganisms produce carotenoids, mainly due to the stressful environmental conditions in which they live. However, not all of them are of industrial relevance [57]. Among the carotenoid-producing microorganisms, bacteria offer several advantages based on short life cycles, metabolic flexibility, and simple propagation techniques. In addition, they can be genetically manipulated [58]. Torularhodin and torulene are widespread microbial carotenoid pigments produced by several genera of microorganisms in high concentrations [59,60].

Microorganisms have developed a complex range of adaptations in their cellular components to grow in different environments, such as maintenance of membrane fluidity, downregulation of flagellar motility, an adaptation of the molecular structure of proteins to ensure greater flexibility at low temperatures, production of intracellular carbon and energy reserves, substances responsible for the absorption of nutrients, synthesis of enzymes involved in the regulation of biosynthetic pathways (biosynthesis of purines and lipids) and degradation of organic compounds, synthesis of substances such as trehalose (natural agent) that confers cryoprotection, and biosynthesis of carotenoid pigments (like canthaxanthin, astaxanthin, and β-carotene, among others). The cell membrane undergoes remodeling of its fluidity as an adaptation to low temperatures, increasing the proportion of unsaturated fatty acids. This modification helps in its semi-fluid state to maintain the function of the outer membrane proteins involved in respiration and transport of nutrients, among other described adaptations [61,62].

Table 2 shows different carotenoids cited in the literature, their sources, and their structural formulas.

*Rhodotorula* species are potent carotenoid-producing microorganisms, specifically beta-carotene, torulene, and torularhodin. Yoo et al. [66] reported the production of a red pigment carotenoid with potential antioxidant and antibacterial activities extracted from the yeast *Rhodotorula mucilaginosa* AY-01. Furthermore, in a recent study, Mussagy et al. [60] provided crucial information for regulatory approval of carotenoid pigment and subsequent commercialization. The authors concluded that microbial torularhodin reduces the risks associated with synthetic dyes, offering greater efficacy and safety for humans, inferring that this xanthophyll can and should be explored in various commercial applications.

### 4.2. Flavins Pigments

Flavins are pteridine-based yellow substances bearing an *N*-heterocyclic isoalloxazine ring (Figure 3). They are produced by plants and most microorganisms [67,68]. Riboflavin (vitamin B2) is a water-soluble compound that exhibits pigment properties. It is the source of all biologically relevant flavins. It originates from flavin mononucleotide (FMN) and flavin adenine dinucleotide (FAD), essential moieties for the activity of flavoproteins and flavo-coenzymes, which play many physiological roles, such as protein folding and metabolism of fatty acids and amino acids [68]. Animals and a few microorganisms do not produce riboflavin and need to obtain it through nutrition as a vitamin [67]. Apart from riboflavin, examples of natural flavins include roseoflavin which is an antimicrobial pigment (Figure 3), a riboflavin analog produced by *Streptomyces* bacteria, and 5-deazaflavins which concern flavin derivatives found in Archaea (e.g., 7,8-Dimethyl-8-hydroxy-5deazaflavin—Figure 3) in which the nitrogen in the five positions of the isoalloxazine ring is replaced by a carbon [67].

Riboflavin is biosynthesized using the purine guanosine triphosphate (GTP) and ribulose-5-phosphate (Ru5P) from the pentose phosphate pathway as precursors [68,69]. GTP provides the pyrimidine portion and the other two nitrogen atoms of the isoalloxazine ring and the ribityl side chain. Ru5P originates the remaining carbon atoms of the heterocyclic ring. The final reaction step, catalyzed by the enzyme riboflavin synthase, consists of a dismutation reaction in which two molecules of the intermediate 6,7-dimethyl-8-ribityllumazine (DrL) exchange four carbon atoms [68,69]. This process is summarized in Figure 4.

Deazaflavins, on the other hand, have diaminouracil and tyrosine as precursors, and their biosynthetic pathway was demonstrated to involve the participation of 5′-deoxyadenosyl radicals to form the typical heterocyclic ring [70].

These flavin pigments are fundamental mediators between two-electron and one-electron processes in biological systems. Indeed, they absorb light, being responsible for displaying a specific color and exhibiting strong absorption in the ultraviolet and visible regions. Riboflavin, commonly known as vitamin B2, is an essential component of living organisms and is the precursor of all biologically important flavins [71]. Microorganisms produce a wide range of secondary metabolites that perform multiple functions for the organisms that produce them, including survival [72].

### 4.3. Tetrapyrrole-Containing Pigments

The functions of tetrapyrroles in microorganisms include light capture and electron transfer reactions. They are cofactors for essential enzymes and sensory proteins. Thus, tetrapyrroles can contribute to oxidative stress and protect cells, contributing to the detoxification of ROS. There is a direct correlation between tetrapyrrole metabolism and oxidative stress and its protection in photosynthetic organisms [73].

Tetrapyrrole compounds, composed of four pyrrole rings connected by methine bridges, are ubiquitous. They constitute the heme group, which is part of hemoglobin, cytochromes, and other proteins, and are also part of chlorophyll and billin pigments [74,75,76]. Tetrapyrrole is biosynthesized from δ-aminolevulinic acid (ALA), a five-carbon amino acid formed by the condensation of glycine and succinyl coenzyme A in animals, fungi, and some bacteria (C4 pathway) or from α-ketoglutarate in plants, algae, archaea, and most bacteria (C5 pathway) [76,77]. The condensation of two ALA molecules catalyzed by porphobilinogen synthase originates porphobilinogen (PBG), with a pyrrole ring in its structure. Porphobilinogen deaminase catalyzes the condensation of four molecules of porphobilinogen, forming a tetrapyrrole ring (uroporphyrinogen III) [74,76], which is further modified and complexed with metal ions to originate compounds such as the heme group, chlorophyll, coenzyme B12 and many others, as shown in Figure 5. In the present section, two groups of tetrapyrrole pigments, chlorophylls, and phycobiliproteins, will be further discussed.

Chlorophylls, the most abundant pigments on earth, are essential for photosynthesis, absorbing light and transducing it into chemical energy [78,79]. They consist of tetrapyrrolic macrocyclic molecules complexed with magnesium. Most are esterified to a long-chain alcohol in C-17 [80]. These pigments absorb light in the violet-blue (400–500 nm) and the yellow-orange/red (600–700 nm) parts of the visible spectrum. Chlorophylls occur together with carotenoids and proteins as light-harvesting complexes in plants, algae, and cyanobacteria. The most common ones are chlorophylls a-d [79]. They differ in the degree of unsaturation of the pyrrolic macrocycle in the side chains, influencing their light absorption properties (Figure 6).

Bacteriochlorophylls are found in anoxygenic photosynthetic bacteria and are related to chlorophylls [81]. They can have shifted absorption bands compared to chlorophyll (under 400 and beyond 700 nm), extending the usable light spectrum. A noteworthy example is bacteriochlorophyll a, the most widely distributed pigment in photosynthetic bacteria, with a purple color [80,81] (Figure 6).

Phycobiliproteins (PBPs) are brilliantly colored water-soluble proteins covalently linked to open-chain tetrapyrrole chromophores known as phycobilins. PBPs are part of light-harvesting complexes in cyanobacteria, red algae, and other algae groups. They are arranged in subcellular structures denominated phycobilisomes, which absorb sunlight from 470–660 nm and transfer the energy to chlorophyll a [82,83]. PBPs are classified into three major groups according to spectral properties: phycoerythrin (red), phycocyanin (blue), and allophycocyanin (bluish-green) [83,84]. Due to essential absorption characteristics, PBPs have emerged as promising fluorescent labeling agents, suitable for applications in fluorescence microscopy, flow cytometry, immunohistochemistry, fluorescence immunoassay, and various other biomedical studies. Red microalgae such as *Rhodella* spp., *Bangia* spp., and *Porphyridium* spp produce red PBPs [85].

PBRs comprise hetero subunits α and β and are commonly found as trimers or hexamers. Each monomer contains two to five phycobilin units bound to cysteine residues [82,83]. Phycobillins are formed from the macrocyclic tetrapyrrole heme by cleavage of a carbon bridge and releasing of an iron atom [86]. There are four types of phycobilins, with different light absorption characteristics: phycocyanobilin (λ_max_ = 640 nm), phycoerythrobilin (λ_max_ = 550 nm), phycourobilin (λ_max_ = 490 nm), and phycobiliviolin (λ_max_ = 590 nm) (Figure 6) [82,86].

Some authors [87] described the strain of *Tolypothrix nodosa* as having the ability to produce toliporphins, which comprise a family of tetrapyrroles. Metagenomic surveys have revealed a diversity of bacteria dominated by Erythrobacteraceae, 97% of which are *Porphyrobacter* species.

### 4.4. Alkaloid Pigments

Some microbial pigments belong to the class of alkaloids. They consist of low molecular weight nitrogen-containing substances in which the nitrogen atom is derived from an amino acid [88]. Several groups of pigments belong to the alkaloid class, such as prodigiosines, tambjamines, and betalains, which will be presented in this topic.

In microorganisms, where some secondary metabolites are generally released into the environment, they have a reserve function. A study where alkaloids derived from polyketides, named citrinin, showed that they are simultaneously synthesized and decomposed during the growth of the fungus *P. citrinum* [89].

#### 4.4.1. Prodigiosines and Tamjamines

Prodiginines are a group of hydrophobic red tripyrrole pigments produced by *Serratia* spp., actinomycetes, and some marine bacteria. The most known pigment of the class is prodigiosin (Figure 7), produced mainly by *Serratia marcescens*. Prodiginines can have a straight chain, such as prodigiosin and undecyl prodigiosin, or can bear a cyclic structure, like cyclononylprodigiosin and butyl-meta-cycloheptylprodiginine (Figure 7) [90,91].

Tambjamines are another class of yellow pigments closely related to the prodiginines. They have a bipyrrole core condensed with a primary amine instead of a monopyrrole. Examples include tambjamine A and C (Figure 7). Tambjamines are found in bacteria (e.g., *Pseudoalteromonas* spp.) and marine invertebrates (e.g., bryozoans, nudibranchs, and ascidians) [92,93,94].

Prodiginines are biosynthesized through a bifurcated pathway that ends up with the enzymatic condensation of the bipyrrole molecule 4-methoxy-2-2′-bipyrrole-5-carbaldehyde (MBC) with a monopyrrole unit, which can be 2-methyl-3-pentylpyrrole, also known as 2-methyl-3-n-amyl-pyrrole (MAP), or 2-undecylpyrrole (Figure 8). MBC is biosynthesized from proline, serine, and malonyl-CoA units. The monopyrrole portions originate from different substrates with different enzymatic apparatuses. MAP has the fatty acid derivative 2-octenal and pyruvate as precursors, while 2-undecylpyrrole is biosynthesized from acetyl-CoA/malonyl-CoA units and glycine [90,95]. The biosynthesis of these pigments is controlled by gene clusters that vary significantly in different bacteria. For instance, in *Serratia* sp., the *Pig* gene cluster is responsible for prodigiosin biosynthesis, while in *Streptomyces* sp., the *Red* gene cluster regulates the process [96].

Kurbanoglu et al. [97] described that although *S. marcescens* is the largest producer of prodigiosin, this pigment is also produced by other bacteria, such as *Streptomyces coelicolor*, *S. lividans*, *Hahella chejuensi*, *Pseudovibrio denitriccans*, *Pseudoalteromonas rubra*, *P. denitrificans*, *Vibrio gazogenes*, *V. psychroerythreus*, *Serratia plymuthica,* and *Zooshikella rubidus*. Indeed, other authors [98] inferred that the biopigment prodigiosin is a red alkaloid dye produced by several microorganisms with a linear tripyrrole chemical structural formula. According to these authors, prodigiosin has a broad spectrum of bioactive factors, such as antibacterial, antifungal, algaecidal, anti-Chagas, antiamoebic, antimalarial, anticancer, antiparasitic, antiviral, and/or immunosuppressive.

#### 4.4.2. Betalains

Betalains are water-soluble *N*-heterocyclic pigments that have betalamic acid as a chromophore. According to their structure and light absorption characteristics, they can be classified into two groups: red-violet betacyanins (λ_max_ = 535–540 nm), formed by condensation of betalamic acid with *cyclo*-DOPA, and yellow betaxanthins (λ_max_ = 460–480 nm), derived from the condensation of the former with various amino acids or other amines [99,100,101]. These pigments are limited in nature and found in plant families of the order Caryophyllales and fungal species of *Amanita* genus [102,103]. Examples include muscapurpurin (purple), muscaurin I, and vulgaxanthine I (orange-yellow) from *Amanita muscaria* (Figure 9) [104].

Betalains are biosynthesized from the aromatic amino acid tyrosine, a shikimate pathway product. The hydroxylation of tyrosine originates *L*-DOPA. This molecule is subsequently converted to betalamic acid under the action of the enzyme 4,5-dioxygenase. The spontaneous condensation of betalamic acid with amino acids or other aminated molecules originates betaxanthins. On the other hand, to form betacyanins, betalamic acid condensates with *cyclo*-DOPA, which also originates from *L*-DOPA through oxidation and cyclization reactions. The resulting molecule, betanidin, is then glycosylated and/or acylated to originate a wide range of betacyanins [101,103,105]. These pathways are summarized in Figure 10.

All members of the betalain family are biosynthesized through the common intermediate betalamic acid, which can spontaneously condense with several primary and secondary amines to produce betalains. In the literature, studies have shown in bioproduction the first complete microbial production of betanin in *Saccharomyces cerevisiae* from glucose. The authors established a way for microbial production of betalains of various colors as a potential alternative to intensive agricultural production [106].

#### 4.4.3. Other Alkaloid Pigments

Phenazines represent a group of aromatic *N*-heterocyclic pigments produced by many bacteria, such as *Pseudomonas* sp., *Nocardia* sp., *Burkholderia* sp., *Streptomyces* sp., and *Vibrio* sp. [107,108]. These compounds, which can present several colors (purple, blue, green, yellow, red, and brown), derive from the shikimate pathway [107,109]. The most known example is the blue phenazine pyocyanin (Figure 11), produced by *Pseudomonas aeruginosa* [108,109].

Violacein is another remarkable example of an alkaloid pigment. This bisindole-violet/blue pigment (Figure 11) is biosynthesized from tryptophan. It is produced by various bacteria genera, such as *Chromobacterium, Collimonas, Duganella, Janthinobacterium*, and *Pseudoalteromonas* [107,110,111].

### 4.5. Polyketide Pigments

Polyketides are a structurally diverse class of compounds formed by the repeated condensation of malonyl-CoA derivatives (acetate/malonate pathway) in a process catalyzed by the enzymatic apparatus denominated polyketide synthases (PKSs). Some microbial pigments belong to this class, as discussed below [112].

#### 4.5.1. Quinones

Some natural pigments assume a quinone (fully conjugated cyclic dione) scaffold, ranging from yellow to red [107]. Some examples are arpink red produced by the fungus *Penicillium oxalicum*, nigrodiquinone A (yellow), obtained from *Nigrospora* sp. fungi, aspergiolide A (red), from *Aspergillus* sp., and bikaverin (red) from *Fusarium* sp. fungi (Figure 12) [113,114,115].

Quinones are derived from oxidizing suitable phenolic compounds [88]. The acetate/mevalonate or shikimate pathways can form them from biosynthesized phenolic systems. The former involves condensing acetyl-CoA/malonyl-CoA units to form a polyketide chain subject to appropriate folding and cyclization, as exemplified in Figure 13 [88,116,117].

#### 4.5.2. Azaphilones

Azaphilone pigments, produced by different fungi genera (e.g., *Monascus* spp, *Aspergillus* sp.), have a typical bicyclic chromophore. Six of them are well known (Figure 14): the yellow pigments monascin and ankaflavin (λ_max_ = 330–450 nm), the orange pigments rubropunctatin and monascorubrin (λ_max_ = 460–480 nm), and the red pigments monascorubramine and rubropunctamine (λ_max_ = 490–530 nm) [118,119,120,121].

These pigments are biosynthesized from precursors from the acetate/malonate pathway. They are proposed to be formed from the esterification of the polyketide-derived azaphilone chromophore with a β-ketoacid. These precursors are biosynthesized under the action of polyketide synthase (PKS) and fatty acid synthase (FAS), respectively [119,120,122], as summarized in Figure 14. The reaction of orange molecules with a primary amine, known as aminophilic reaction, would lead to red substances in which the heterocyclic oxygen is replaced by a nitrogen atom [119].

Five main biosynthetic pathways for azaphilones, controlled by different genes/enzymes, have been identified. In all of them, a PKS builds a common intermediate further tailored by different enzymes. For *Monascus* sp., for instance, seven azaphilone biosynthetic gene clusters have been described [121].

### 4.6. Phenol-Containing-Pigments

A remarkable group of polyphenolic pigments are the styrylpyrones [1]. These yellow compounds are mainly found in fungi from the Hymenochaetaceae family, which includes the genus *Phellinus* and *Inonotus*, and can also be encountered in some plant families (e.g., Lauraceae and Ranuculaceae) [123]. The first styrylpyrone, hispidin, was isolated in 1889 from *Inonotus hispidus.* Many other styrylpyrones can be obtained from its fruiting bodies, such as bisnoryangonin and hypholomin B which is a dimer (Figure 15) [123,124]. Hispidin is a biosynthetic precursor of fungal luciferin in bioluminescent fungi of the order Agaricales [125].

Studies evidenced that the styrylpyrone scaffold is biosynthesized with phenylpropanoid units from the shikimate pathway and malonate units (acetate/malonate pathway) under the action of type I polyketide synthases [123,125]. The proposed biosynthesis of hispidin is shown in Figure 16. More complex dimeric and polymeric styrylpyrones are likely formed through an oxidative coupling of monomeric styrylpyrone building blocks [123].

Polyphenol-containing pigments extracted from genetically modified microorganisms exhibit important antioxidant activities. We can cite the flavonoids, nagirenim, and anthocyanin from *Saccharomyces cerevisiae* through metabolic engineering [126] and curcuminoid from engineered *E. coli* [127,128].

### 4.7. Melanins

Melanins are ubiquitous high molecular-weight pigments formed by the oxidative polymerization of phenolic compounds. They are stable, insoluble, and usually brown or black [129,130,131]. Three major types of melanins are recognized: eumelanins, pheomelanins, and allomelanins. Eumelanins are found in animals, fungi, and some microorganisms. They are biosynthesized from tyrosine with the participation of the enzyme tyrosinase and have 5,6-dihydroxyindole as the main building block. Pheomelanins, present in higher animals (mammals, birds, and reptiles), differ from eumelanins by sulfur, incorporated from *L*-cysteine to the tyrosine-derived units, forming benzothiazine and benzothiazole building blocks. Allomelanins, on the other hand, consist of a heterogeneous group found in plants and fungi. These melanins are devoid of nitrogen atoms and are formed from different nitrogen-free precursors, such as catechol, which is the most common, and gamma-glutaminyl-3,4-dihydroxybenzene 1,8-dihydroxy naphthalene, as well as caffeic, chlorogenic, protocatechuic, and gallic acids [129,131]. The biosynthesis of these groups is summarized in Figure 17.

The biosynthesis of melanin in several fungi employs hydroxynaphthalene subunits, formed from the acetate/malonate pathway under the action of PKSs, a process controlled by different gene clusters. In *Aspergillus fumigatus*, for instance, a cluster formed by six genes encompassing 19 kb pairs of DNA has been described, and a similar cluster was detected in *Penicillium marneffei* [132,133].

Rao et al. [134] described the production of different types of melanins by a wide variety of microorganisms, such as *Colletotrichum lagenarium*, *Magnaporthe grisea*, *Cryptococcus neoformans*, *Paracoccidioides brasiliensis*, *Sporothrix schenckii*, *Aspergillus fumigates*, *Vibrio cholerae*, *Shewanella colwelliana*, *Alteromonas nigrifaciens,* and many species of the genus *Streptomyces*. Furthermore, a recent work described the remarkable production of melanins in bacteria isolated from sponges, suggesting that these microorganisms are potential sources for industrial melanin production [135]. These results suggest that bioprospecting for melanin-producing bacteria, including those associated with other organisms or living in extreme environments, could help find melanin hyperproducers or extremozymes that could increase melanin yields. 

Table 3 summarizes other biopigments found in microorganisms, their biological activity, and their commercialization status.

Molelekoa et al. [151] characterized different microbial pigments extracted from fungi, such as bostrycoidin (a red aza-anthraquinone), with antioxidant and antimicrobial properties produced by *Fusarium solani*. Another red pigment called rubropunctamine extracted from *Talaromyces verruculosus* with antioxidant and antimicrobial biological activities is described too, alongside sclerotin (yellow-brown) with antioxidant, antibiofilm, and antimicrobial properties produced by *Penicillium multicolor* and *P. mallochi*. Some microorganisms can grow under different physicochemical conditions due to metabolites like carotenoids. In another work, the authors suggested that staphyloxanthin, a golden carotenoid biopigment extracted from *Staphylococcus aureus*, is an important virulence factor due to its antioxidant properties [151].

## 5. Challenges of the Industrial Application

Despite the great potential for industrial use, microbial pigment production and commercialization present some challenges. The resistance is mainly due to production costs, despite being an innovative alternative, opening a new panorama of sustainable and green products that bring gains to the environment and human and animal health. Genetic modifications for better production, optimizations of the bioprocess, improvement of the extraction methods, and use of technologies such as microencapsulation, among other approaches are studied.

### 5.1. Converging Frontiers: Exploring the Synergy of Multiomic Integration, Synthetic Biology, Artificial Intelligence, and Metabolic Engineering

Due to the importance of microbial pigments and their industrial application, synthetic biology and metabolic engineering strategies have been developed to optimize yield production and reduce bioprocess costs. For this purpose, different strategies are used to build new biosynthetic pathways by recombining multiple target genes or regulating crucial molecule precursors in the biochemical chain to enhance the final yield of a bioproduct. They appear to be essential tools for the future market of these bioproducts [152,153]. The possibility of genetic engineering expands the perspectives of the production and application of microbial pigments, which can have their process optimized. These overexpressed genes regulate pigment production, growth, and fermentation in different culture conditions, as well as the cloning of genes in different microbial species more adapted to industrial processes, among other possibilities [154].

Science fields, including microbiology, were strengthened in the 2000s after multi-omic (genomics, transcriptomics, metabolomics, proteomics) approaches were developed. In microbial biotechnology, they provided a rapid expansion of knowledge of molecular biology and metabolic pathways in microorganisms and the possibility of gene editing in microorganisms for specific applications [152,155]. Moreover, artificial intelligence can be used with omics sciences for better investigation and gene assembly of microbial agents in biotechnology. In this scenario, multi-omics can provide microalgae data to be inputted into computational systems biology and machine learning (heatmap analysis, clustering, neural network, and others), followed by a guided confirmation in the laboratory once target genes and metabolic pathways are identified. By combining multi-omics and predictive microbiology mathematical models, for example, it is possible to predict microbial growth or optimize culturing processes involved in their bioproduct production, including pigments [156].

In this context, metabolomic was an important tool used in a work that successfully studied an enhanced production of lutein carotenoid by the *Chlorella saccharophila* strain in co-cultivation with a novel *Exiguobacterium* sp. [157]. In this work, metabolomics permitted authors to identify metabolites from microalgae and bacteria cultures alone and together, in addition to observing more than 40 metabolites unique when in co-culture. Another work used multi-omic tools to study the production of carotenoids in *Isochrysis galbana* microalgae and the genes involved under different light conditions during growth [158].

Synthetic biology is a well-known tool in the biotechnology of pigments and their microbial producers due to the possibility of controlling and improving the biosynthesis of a bioproduct through modification in microbial genes. Researchers construct heterogeneous synthesis pathways after identifying functional genes for pigment production chains by direct cloning or through a metagenomics tool, followed by pathway assembly targeting the recombination of target genes (plasmid or plasmid-free techniques) [144]. Following this concept, an increased yield of β-carotene was obtained after expressing three lipase and carotene pathway encoding genes from other microbial genera into the baker yeast *Saccharomyces cerevisiae* [159]. In a combinatorial multi-gene pathway assembly, lycopene pigment production was increased three times in an expression in *E. coli* [160]. Lycopene production was also incremented in another work that used recombinant genes in engineered yeasts (*S. cerevisiae*) to optimize lipid production to pigment accumulation reaching up to 73.3 mg/cell dry weight [161].

Biochemical engineering technology was used to regulate the biosynthesis of pigments from a mutant strain of the filamentous fungi *Monascus purpureus* LQ-6, which produces several bioactive compounds [162]. This technology consists of adding exogenous compounds in fermentation to alter cell metabolism, thus easing the regulation of target bioproducts. The authors observed that by adding 1.0 mg/L of methyl viologen and rotenone as cofactor agents, total yield production of pigment increased by approximately 40% and yellow pigment production by around 114%, respectively. The same work also showed that pigments changed from red to yellow through an electrolytic stimulation with no citrinin which is a mycotoxin produced by these strains. Other metabolic engineering strategies, such as extractive fermentation [163] and low-frequency magnetic field exposure [164], were also investigated for pigment production in *Monascus* spp. strains. These examples and others found in the scientific literature show that the future of metabolic engineering is a promising tool for pigment development studies.

Another strategy of metabolic engineering is the use of CRISPR/Cas technology. The pigment genes could be introduced into the bacterial gene using the CRISPR-Cas9 system to engineer and produce natural pigments. It can be used in a wide variety of bacterial cells, such as *Corynebacterium, E. coli*, *Pseudomonas*, *Staphylococcus*, *Bacillus*, *Clostridium* sp., *Lactobacillus* sp., *Mycobacterium* sp., and *Streptomyces* sp. [165].

### 5.2. Fermentation Process

Optimizing fermentation processes makes the bioprocess more profitable and is usually the key for an industrial application of bioproducts. They can come from modifications in the controlled cultivation process, but they can also be more punctual and specific [166]. With research and investments in biotechnology, genetic modifications of microorganisms are feasible to increase pigment production or affect other parallel characteristics that also help manufacture colors [43,167]. It is necessary to consider that the world’s bioprocess demand has been increasing in the last five years. The bioreactors market was valued at $2615.4 million in 2020 and is estimated to reach $7328.4 million by 2030, growing at a CAGR of 10.7% from 2021 to 2030. Development in the biotechnology and pharmaceutical industry and advancement in R&D for cell culture and microbial application are the major factors that drive the growth of the global bioreactors market [168]. In this context, improvements in bioprocesses for microbial pigments are in progress and are estimated to reduce costs.

Scaling up the production of microbial pigments is a bottleneck. Fermentation costs and low production yield are issues that limit the expansion of pigment application widely in industrial scenarios. Nevertheless, technological advancements have developed bioprocess solutions such as fermentation optimization and the use of agro-industrial wastes. An economically and industrially viable production process for microbial pigments has been investigated through fermentation tanks for high-yield production, applying metabolic engineering as previously described, including experimental planning statistics and different fermentation techniques, such as cell-immobilization, to improve pigment production [156,169].

Several microorganisms are known for growth suitability in culture media with alternative nitrogen and carbon sources provided by agro-industrial wastes and by-products [170,171]. An extensive and recent review by Rocha Balbino et al. [172] discussed how millions of tons of lignocellulosic biomass are discarded per year due to agro-industrial activity. Microorganisms can use this material to produce different bioproducts, such as biofuels, enzymes, and pigments. Microorganisms can also be agents for natural pigment extraction from these agro-industrial wastes through fermentation [173]. This practice has been very sought after in recent years because it stimulates sustainability concepts in industrial processes and reduces bioprocess costs.

Following this idea, carotenoids from *R. mucilaginosa*, *Sporidiobolus pararoseus,* and *P. fermentans* yeasts were produced in agro-industrial media (raw glycerol, corn steep liquor, and sugarcane molasses) presenting good antioxidant activity and promising industrial application [174]. Similarly, carotenoids from *Rhodosporidium toruloides* were produced with yields of 3.6 mg/g agro-waste sugars from wheat straw hydrolysate [175]. In another work, submerged and solid-state fermentation strategies were successfully developed for pigment production by *P. brevicompactum* using cheese and corn-step liquor industrial wastes [176]. Thevarajah et al. [177] elaborated an extensive and enlightening review of pigments from cyanobacteria (namely *Arthospira* spp., *Nostoc* spp., *Anabaena* spp., and others) which were produced in wastewater media as an alternative energy source. A review regarding microbial pigments produced by using agro-industrial wastes was published by Panesar and colleagues [48]. Other works that describe microbial pigment production using agro-industrial wastes or other alternative substrates in fermentation can be found in Table 4.

Together with the utilization of alternative substrates, optimization of pigment production from various microorganisms has been investigated through statistical analysis since it is a powerful tool for identifying optimum conditions for microbial cultivation and pigment biosynthesis during fermentation processes. In the last years, several works utilized different experimental planning for studying (namely, Placket-Burmann, fractional factorial, central composite design of response surface methodology, and others) pigment production by bacteria, fungi, and microalgae [185,186,187,188,189].

Additionally, studies in cell immobilization techniques have been investigated to optimize pigment production bioprocesses alongside the methods described previously. For example, a study reported significant carotenoid production by a *Planococcus* sp. bacterial strain when immobilized in a waste-based matrix generated from the pulp and paper industry [190]. The same work studied the effluent treatment during the bioprocess, in addition to pigment production and agro-waste utilization, simultaneously solving multiple industrial and environmental issues. Carotenoid production from *Chlorella minutissima* immobilized in alginate increased by 32.6% (11.0 mg/g^−1^) compared to cell-free fermentation [191]. Cell-immobilization technology also enhanced the production of yellow pigment from *M. purpureus* under submerged fermentation in a recent work [192]. According to the authors, pigment production was improved due to the immobilization of the cells and the addition of sucrose esters in repeated batches, reaching a maximum of 56.71 AU/50 mL of pigment. *M. purpureus* strains were used in previous works for pigment production with immobilization techniques [193]. These findings unlock opportunities for different strategies and technologies for optimizing pigment production from different microorganisms on a pilot and industrial scale.

### 5.3. Pigment Extraction

Another stage that needs concern in industrial-scale processes involves pigment extraction techniques. Once microbial pigment producers are found, researchers study production followed by efficient extraction of pigments before investigating potential applications for it. Downstream engineering processes, including bioproduct extraction, usually comprise more than 50% of the total cost of bioprocesses. Once there is no unique way to extract pigments from microbial cells, it is essential to investigate different methods to find the most feasible methodology to obtain the bioproduct with the least cost. Moreover, the cleaner and “greener” these processes are regarding environmental subjects, the better for a promising industrial application [150,194,195,196].

In extraction processes, several factors may influence the microbial pigment yield obtainment, such as solvent choice, type of pigment, complexity of microbial cell wall, and presence of culture media residues [195]. Conventional extraction of different pigment groups may include maceration, distillations, Soxhlet extraction, and water/solvent infusions [150]. Even being used until present days, it is well-known that these methods present disadvantages, such as large volumes of solvent usage, poor extraction, time-consuming processes, chemical degradation of pigment structure when performed with high temperatures, and thus loss of bioactivity [37,150,197]. Moreover, during these processes, the use of organic solvents such as acetone, benzene, petroleum ether, hexane, and methanol is prevalent. These solvents are toxic, highly inflammable, and non-biodegradable [198]. Hence, due to sustainability and green technology approaches, pigment extraction without using organic solvents other than ethanol or water is undesirable [38].

Therefore, novel and advanced techniques have been developed to extract pigments that diminish environmental hazards compared to conventional methods. Extraction techniques such as electric-pulsed, enzymatic, ultrasound, and microwave-assisted, among others, are described as newer and feasible methods for obtaining melanin, carotenoids, and chlorophylls [199,200]. These techniques usually require less procedure time, with low to average consummation of solvents (water, aqueous, and non-aqueous), in addition to being highly efficient and financially viable for industrial-scale extraction. Conversely, they may need high pressure and temperature during the process [197,201].

In a recent study, carotenoid from *R. mucilaginosa* was better extracted using methanol and an ultrasonication system [181]. Pulsed-electric field and ohmic heating have been used together with solvents in attempts to enhance pigment recovery yield from microalgae [201,202,203,204]. Pigment extraction based on adsorption chromatography with resins has also been studied, showing a promising result for industrial application once it could reduce costs and eliminate the required steps of conventional extraction methods [37]. Prodigiosin extraction from *S. marcescens* [205] was studied following this technique. However, it has also been studied for pigment extraction from sources other than microbial [206,207]. Few scientific articles report enzyme-assisted extraction methods for microbial pigments, suggesting an area to be explored in future research. One described different enzymatic techniques for cell lysis of *H. pluvialis* microalgae leading to carotenoid extraction. Almost 84% efficiency was achieved by combining this method with the ultrasound technique [208]. Based on these successful studies of non-conventional methods for microbial pigment extraction, it is possible to envisage a more extensive application of these techniques in the industrial scale of the sector in the future.

### 5.4. Micro and Nanoencapsulation

Microbial and natural pigments suffer from degradation when exposed to some physical and chemical agents, such as light, oxygen, metals, acids, moisture, and high temperatures in any stage of processing or storage for application. This degradation occurs through chemical alterations (oxidation, isomerization) in pigment structure, which may result in loss of the bioactivity properties, color fading, release of undesirable odors, limited shelf-life, and thus as a consequence impacting negatively on the quality of the final product. [209].

Pigment encapsulation appears to preserve the integrity of pigment molecules and their properties. Generally, it consists of incorporating these biomolecules into some colloidal particle, which ranges from micro (100 nm–1000 μm) to nanoparticles (1–100 nm) [209,210]. When encapsulated, pigments can resist the conditions of the surrounding environment since they are physically isolated. Additionally, their functionality can be improved once colloidal particles used for pigments can be designed to change their properties in specific environmental triggers (pH, temperature, enzyme activity, and others), acting as delivery-system tools for a specific application of these pigments. Depending on the desired function, they can enhance dispersibility and solubility in water or lipid pigments [209]. For encapsulation, some factors are considered criteria for particle materials, including biodegradability, resistance, and low hygroscopicity. Pigments are incorporated in the micro or nanoparticles through spray-drying, fluid-bed coating, freeze-drying, and emulsification [39,164].

The importance of encapsulation properties can be demonstrated in a recent work where rubropunctatin from *Monascus* sp. was involved in nanoparticles of lecithin/chitosan from 110–120 nm size [211]. These particles possessed excellent solubility in water and dispersibility and presented an anti-tumor effect against target cells through delivery-system release. Another recent work also showed the stability and extended shelf-life of *Monascus* sp. pigments with nanoparticles of zein-lecithin [212]. From a different point of view, both micro and nanoparticle chitosan-based delivery systems of prodigiosin (*S. marcescens*) were prepared with promising results in-vitro and in-vivo assays in the animal system [213]. With these findings, various industrial sectors can gain innovations in applying these micro and nanoparticles.

Numerous reports can be found in the scientific literature regarding the encapsulation of microbial pigment with an industrial view. A microencapsulated canthaxanthin carotenoid from *Dietzia natronolimnaea* bacteria was developed in Alginate—High methoxyl pectin with 70% method efficiency, maintaining its antioxidant property in both neutral and acidic environments [214]. These results showed a promising application of these encapsulated pigments in dairy products. A microparticle violet colorant consisting of pigment from *C. violaceum* was prepared using Arabic gum and presented satisfying storage stabilities for application in food products [215]. Micro and nanoencapsulation technology allows for the integration of microbial pigments in the food industry, which has been gradually present in the last years [216].

Focusing on cosmetic sector application, pigments from *T. australis* and *P. murcianum* fungi strains were successfully microencapsulated in a biopolymers blend (sodium alginate, chitosan, and hyaluronic acid) particle with roughly 500 μm and 30–40% method efficiency. Microcapsules exhibited antioxidant activity and presented an acceptable powder form for cosmetic formulations [217]. Melanin nano-encapsulated pigments have also been studied for application in cosmetics and biomedical industries due to their antioxidant and photoprotective properties [199,218]. As an example, a protective activity against ultraviolet radiation of nano-based sunscreen formulation of melanin produced by *Pseudomonas* sp. was described [219].

In an innovative study, astaxanthin from *H. pluvialis* was obtained by enzymatic lysis assisted by ultrasound and then microencapsulated in spherical particles of co-polymer with approximately 0.2 μm at an efficiency rate of 51% using a high biomass/dichloromethane ratio (10 mg/mL) and a low precipitation pressure (80 bar) [208]. The last-mentioned work has used welcoming steps and technologies for the industrial application of pigment in the future, with efficient and environmentally friendly methods. These promising findings encourage scientific and industrial bodies to invest in this downstream stage of pigment chain production to better the effects of their application on a larger scale.

## 6. Major Biotechnological Applications

### 6.1. Pharmaceutical and Medicine

Pharmaceutical and medicinal industries use pigments from different sources, such as microbial or higher algae and plants, because of their bioactive properties for adding value to the products. Pigments can be used as antimicrobial metabolites; many of the pigments already presented have antibacterial and antifungal activities. This property can benefit pharmaceutical and cosmetics formulations, increasing the safety of formulations and especially the shelf lives of the most perishable products [3]. The antimicrobial activity includes viruses (e.g., prodiginines), bacteria (e.g., pyocyanin), and fungi (e.g., violacein) [137]. Several microbial pigments, such as melanin, prodigiosin, violacein, and others, present anticancer, anti-inflammatory, and antioxidant properties as described [151,220,221,222]. As demanded by pharmaceutical industries, studies have been conducted regarding microbial metabolites to discover new bioactive compounds, including pigments. Prodigiosin from *S. marcescens* is well-known for its bioactive properties, and it has been described as a drug candidate for therapy in cancer and neurodegenerative diseases [223]. More specifically, prodigiosin has already been tested in different concentrations against different tumor cell lines and presented encouraging results, such as for pancreatic cancer [224], epithelial carcinoma [91], breast cancer, lung cancer [98], and others. Violacein extracted from *C. violaceum* has also been described as an antitumor molecule, promoting the death of osteosarcoma and rhabdomyosarcoma cells [225]. In addition, it is also known to be anti-inflammatory through mechanisms that reduce pro-inflammatory cytokines and chemokines, such as TNF-α, TGF-ß, IL-1β, IL-6, CXCL1, and CXCL12. Also, it promotes other factors for anti-inflammatory drugs such as interleukins IL-4 and IL-10 [226]. More primary in vitro studies indicate that 1 mM violacein, extracted from *Janthinobacterium* sp., inhibited 94.3% of HIV-1 reverse transcriptase, and 2 mM inhibited the binding between the SARS-CoV-2 spike glycoprotein and the ACE2 receptor by 53% [227]. Other studies associated with violacein demonstrate antifungal potential similar to those obtained with bavistin, used as a fungicide in agriculture, and with amphotericin B, a drug already used in the treatment of mycoses, such as candidiasis and mucormycosis [228,229,230].

Recently, authors have described sclerotiorin, rubropunctamine, and bostrycoidin pigments from *Penicillium multicolor, Talaromyces verruculosus*, and *Fusarium solani*, respectively, presented antioxidant and antimicrobial activities against Gram-negative and Gram-positive bacteria [151]. An anthraquinone pigment produced by *T. purpureogenus* was purified and exhibited antioxidant activity and anticancer properties against tumor cell lines. It has been pointed out as a potent agent for kidney cancer diagnosis [136]. Antibiofilm property was also described in a study investigating the therapeutic characteristics of a pigment from *P. mallochi* ARA1 [231]. The development and studies of novel natural pigment compounds are continuous in the scientific literature and are a broad field to explore.

The aquatic environment has also been a source of studies to discover new microbial species and new molecules. Notably, deeper marine environments present even more exciting characteristics, as they tend to have resident microbes that live in more different conditions, for example, acidic, alkaline, with higher salinity, low availability of nutrients, total or partial deprivation of sunlight, and high and low temperatures, among others. Therefore, it is understood that metabolites produced under these conditions may be of better industrial interest due to the possibility of being more stable and durable [228]. A new species of *Micrococcus* producing a yellow pigment not identified by this study was isolated in the Persian Gulf. This pigment demonstrated antibacterial activity against *P. aeruginosa*, *E. coli*, and *S. aureus* [232]. Within the marine universe, a strain of *Streptomyces* sp. melanin producer was able to inhibit the growth of other species such as *Klebsiella pneumoniae* sp., *B. subtilis* sp., *Proteus vulgaris* sp., and *P. aeruginosa* sp. [233,234].

### 6.2. Cosmetics

Artificial or natural inorganic and organic pigments are primary sources of coloration in the cosmetics industry. Nevertheless, bio-based colorants are sought for sustainable concepts and additive functions such as antimicrobial and antioxidant activities in these bioproducts [235]. Plant pigments are well-known sources for the cosmetics industry [235,236,237].

Regarding natural sources from microbial origin, microalgae and cyanobacteria are producers of high-value pigments such as carotenoids, phycocyanins, and chlorophylls for developing product colors with tons of yellow, orange, red, green, and blue, for example [152,238,239]. The application of microalgae for the production of pigments and other bioproducts, such as vitamins and lipids, has been supported by scientists due to their possible integration into biorefinery and sustainable production in the industry [240]. Various genera of bacteria and fungi have been described and studied for melanin production, which can be applied in the cosmetics industry as an ingredient in dermal products, such as sunscreen, due to high UV light protection properties. This property of protecting UV radiation in producing microbial cells can be extended to cosmetics when this pigment is extracted and included in a cosmetic formulation with photoprotective intent with the safe possibility of human use. Some scientific works already describe this use, where rubropunctamine and monascin, both derived from the fungus *M. purpureus*, were added to sunscreens and enhanced their photoprotective action, increasing this effect by approximately 36.5% and 13%, respectively [241,242]. Studies also tested melanin extracted from fungi as an additive in sunscreens and obtained favorable results in their analysis, demonstrating the potential of microbial pigments as enhancers of sun protection factors [243].

The vast bioactive properties and different types of microbial pigments demonstrate the potential of microbial pigments to be applied to the cosmetics industry [244], such as sunscreen, makeup, antiaging, skin lightening, and even for tattoos and permanent dyes combining the biological activity and coloring property. These applications are not yet widely explored by the industry. However, it is a promising appeal for this sector [169,245].

Zoz et al. [61] described that these pigments had pro-vitamin A activity and potential application as coloring additives in foods and cosmetics, like the lycopene carotenoid pigment, due to their red color.

Among the various sources of melanin, fungal melanin is a promising candidate for sunscreen due to its sustainability and scalability. Jeong-Joo Oh et al. [246] studied melanin samples derived from *Amorphotheca resine.* The authors evaluated sunscreen performance and found higher antioxidant activity than conventional antioxidants. The pigment would cause no cytotoxicity towards human keratinocytes. Consequently, they concluded that fungal melanin could be utilized as a multi-purpose broad-spectrum sunscreen agent with no cytotoxicity, contributing to replacing or diminishing some synthetic broad-spectrum sunscreens in cosmetic formulations [246].

### 6.3. Food Industry

The food industry is notoriously the industrial field with the highest application of the use of microbial pigments. Pigments may be one of the most critical components in food because food color is an essential sensorial characteristic for consumer acceptance. In addition to the sensorial aspect, natural pigments have been preferred in the food industry due to their potential health effects, in contrast to artificial colorants, as well as a conserving and shelf-life extending agent due to their antioxidant properties [247]. Natural pigments already represent one-third of the pigment additive market in the food production sector, mainly in the carotenoid class, whose prominent representatives are β-carotene and astaxanthin. As seen in Table 2, numerous microbial species can produce both and are already a reality in industrial food applications [248].

Some pigments, namely riboflavin (from *B. subtilis*), β-carotene (from *Blaskslea trispora* and *Dunaliella salina*), lycopene (*B. trispora*), and astaxanthin and canthaxanthin (*Haematococcus lacustris*) have been already extracted and commercialized in the current global market [249].

Astaxanthin is an antioxidant that could be used in animal or human food for promoting color [250]. The use of microbial pigments in animal husbandry is an interesting application that has gained attention in recent years. Microbial pigments can be incorporated into animal feed to enhance the coloration of animal products such as eggs, poultry, and fish, making them more visually appealing to consumers. For example, astaxanthin, a red microbial pigment from the microalgae *Haematococcus pluvialis*, is often used to improve the coloration of salmon, shrimp, and egg yolk. It is used in a preparation called the Red Meal (RM), which is the whole cracked biomass of *Haematococcus pluvialis* homogenized in vegetable oil [251]. It is important to use carotenoids in fish feeds. These give yellow, red, and pink to fish skin, flesh, and eggs [252]. A work observed greater pigmentation of koi carp fish (*Cyprinus carpio*) after being fed with a diet of marine yeast *Rhodotorula paludigena* VA242 which is a producer of carotenoids [253]. Another application is the use of astaxanthin to maintain the color and oxidative stability of raw ground pork meat during refrigerated storage [254]. In some cases, astaxanthin supplementation can even accelerate the animal’s growth and, as in the case of trout (*Oncorhynchus mykiss*), optimize the production chain for the food industry [254].

Microbial canthaxanthin is used as a pigment additive for poultry and salmon feed. Canthaxanthin can be obtained through the bacteria *Bradyrhizobium* sp., a photosynthetic bacterium, and *Lactobacillus pluvalis* [3,250,254]. Aquafeed has used canthaxanthin for years to impart the desired flesh color in a farmed salmonid. *Bradyrhizobium* sp. canthaxanthin could be used in this segment.

Supplementation in animal feed is usually used because canthaxanthin in the ration contributes to the pigmentation of animal muscle tissue, as in the case of salmon and shrimps, in the production of eggs in chickens, and in obtaining more vivid colors in birds and canaries. Furthermore, canthaxanthin and other carotenoids are antioxidants that can help control free radicals and fertility in these animals [255,256,257].

Chlorophylls from several microalgae origins (*Chlorella* spp., *Tetraselmis* spp., *Arthospira* spp.) are widely used in the food industry for green colorants in addition to their bioactive properties, including anticancer, for example [258]. For instance, a colored functional oil rich in fatty acids and antioxidants from *C. vulgaris* microalgae was developed using less environment-harmful technology (supercritical CO_2_ extraction) [259]. Similarly, a yellowish pigment from *M. purpureus* H14 fungi was studied for application in a functional rice noodle since it provided more stability, quality, and texture to the product apart from its bioactive properties (anti-inflammatory, anti-tumor, anti-fatty) [260].

Other microorganisms, such as fungi, are widely used in the food industry. The fungus *Monascus* sp. is a genre historically used by Asian people to ferment rice, creating a culinary delicacy called “red mold rice”, which is even used as a functional food, providing color flavor and increasing the food’s shelf life [261]. Variations of this rice have been described as safer when fermented with the species *M. purpureus*, due to the lower possibility of the presence of mycotoxins. Other fungi are also present in fermentation processes to promote flavor and color in foods, such as *P. roqueforti* for Roquefort cheese/Blue cheese and *P. camembert* for Brie cheese, among others [262,263]. Another example is *Blackslea trispora*, a fungus that produces β-carotene, which is approved for food use in the European Union [264].

Applications of microbial pigments in food must be approved and meet the requirements of regulation and legislation institutions, such as the GRAS (Generally Regarded as Safe) label and the Food and Drug Administration (FDA, Silver Spring, Maryland, USA), and some microbial producers have already taken this action [258,265]. To conclude, even with some usage limitations (instability and costs, for example), microbial pigments are in high demand for the food market. They are trending to increase production and application in future years [265].

In some cases, new technologies are used. They are constantly improved to increase the action and conservation of these microbial pigments, which are sometimes more sensitive to environmental conditions than synthetic pigments. Therefore, microencapsulation and nanoencapsulation can protect microbial pigments from environmental factors, such as heat, oxygen, humidity, direct incidence of UV radiation, light, and others, to extend their useful life. There is not only the possibility of masking unwanted flavors and odors that may be associated with the pigments obtained. Furthermore, it is seen that some pigments of a hydrophobic nature, such as carotenoids, may have improved solubility in water when encapsulated with appropriate wall materials, such as modified starch or maltodextrin, and cases of use of these are already underway in the coloring industry of yogurts, soft drinks, and cakes [266,267].

### 6.4. Textile Industry

Applying microbial pigments for dyeing fabrics is not a common practice in industries. It has still been little explored in scientific research. However, some more recent and initial work shows the use of these pigments in different fabric types. They are divided into three stages: preparation of the dyeing solution containing the microbial pigment color fixing additives, hot dyeing (60–80 °C), and washing and drying the dyed fabric [268].

Violacein pigment with bioactive properties (antimicrobial and antioxidant) was investigated for application in textile purposes [269]. The authors extracted violet/purple pigment from the bacterial producer *C. violaceum* to dye cotton and silk. Good results were obtained on a small scale with color stability after the dyeing, washing, and drying. Furthermore, hot-dyed strips of cotton fabric with dark green pigment extracted from the fungus of the genus *Sclerotinia* were prepared after successive washing and exposure to the sun [270].

Bisht et al. [271] proposed using antimicrobial pigments to develop an antimicrobial textile fabric in the textile industry. A crude extract of red pigments from *Rhodonellum psychrophilum* was applied on cotton, rayon, and silk fabrics. Penetration and fixation of the pigment were observed, but without maintaining vibrant color. Likewise, a red pigment produced by *Talaromyces albobiverticillius* fungi in fermentation using agro-industrial waste was studied for dyeing cotton fabric and presented antimicrobial and antioxidant properties, showing a promising path of developing textile materials with bioactive properties [182].

*Scytalidium cuboideum* and *S. ganodermophthorum* are fungi species with red and yellow pigments, respectively. They can be used in dyeing cotton, nylon, and polyester fabrics [272]. Another possible fungus is *Cuvularia lunata*, a producer of melanin used for dyeing silk and wool and red pigments derived from *Isaria* sp., *Emericella* sp., *Penicillium* sp., *Monascus* sp., and *Fusarium* sp. for dyeing leather.

Chadni et al. [273] isolated a strain of *Talaromyces verruculosus*, producing a red pigment which was extracted and used together with a fabric mordant to dye cotton fabric and maintain the durability of the pigment in this fabric. This study found that the dyeing was satisfactory, and the color remained stable after washing. In addition, a cytotoxicity test was carried out, which confirmed that the human use of fabrics dyed with the *T. verruculosus* pigment was safe [273].

Suwannarach et al. [274] isolated a strain of *Nigrospora aurantiaca* from the plant *Cinnamomum zeylanicum.* This fungus produces a red pigment, characterized as bostrycin, which has also been tested for dyeing in different types of fabrics. The procedure was performed in five consecutive dyeings with the same pigment to ensure better impregnation of the pigment into the fabric. In the end, a reddish tone was presented in the silk, cotton, and nylon fabrics, the cellulose fabric was pale red, and the acetate fabric was pink. However, both polyester and wool fabrics showed pinkish coloration. In this study, all fabrics lost a little color after washing, maintaining their original color slightly lighter after drying.

Not only are pigments derived from fungi and bacteria undergoing scientific advancement in the textile industry, but microalgae also show potential for such use. Mutaf-Killic et al. [275] bring chlorophylls extracted from *Caulerpa lentillifera* for silk dyeing. Furthermore, this work presents challenges that need to be overcome to popularize microbial pigments for industrial dyeings, such as high financial investment, improvements in extraction processes, yield, and color stability. However, despite all this, it is an expanding market, rich in inventive and innovative possibilities, with more sustainable products for the planet, thus bringing another door to biotechnological solutions.

## 7. Pigment Market

As extensively discussed in this present review and found in critic and review papers in the scientific literature, microbial pigments are highly promising for industrial applications and compete with synthetic pigments [276]. According to a publication by Allied Market Research in 2023, the pigments market has been growing, valued at 1.8 billion dollars in 2021, considering the carotenoids segment alone. It is projected to reach $2.7 billion by 2031, growing at a CAGR of 3.9% from 2022 to 2031. Inside this market, Europe and the US are more focused on natural pigment additives to reduce health impacts in the population that consumes artificially colored foods. In addition, FDA (Food and Drug Administration) has already approved the consumption and commercialization of some pigments derived from plants and microorganisms for human consumption as food additives, such as Arpink Red (*Penicillium oxalicum*) and astaxanthin (*Xanthophyllomyces dendrorhous*) [253,254,255,256].

One of the significant advantages of microbial metabolites is their suitability in high demand for “natural”, “biodegradable”, and “eco-friendly”, for instance. These possibilities help construct a good image and status for the brands and industries that use them, influencing the consumer to trust the product. Marketing for natural product consumption supports advertising campaigns that constantly expand this market, pushing it for innovations to reach more and more market share. Recent data shows that the global organic pigment market is expected to generate around 4.89 billion USD by 2024 [231,248].

Among the microbial pigments, the most applied in the industry are those extracted from microalgae and fungi, with less use of those from the bacterial group [248,277].

Pigments that bring antioxidant characteristics, for example, can increase the product’s shelf life or even reduce the addition of other compounds commonly added in formulations. These gains must be analyzed case-by-case for each formulation and pigment; however, this opens a new horizon in the study, production, and use of microbial metabolites in different industry sectors for replacing traditional dyes and pigments [3].

Carotenoids already have an expanding place in food coloring in the food industry. However, its vibrant colors vary in scales from yellow to red, depending on the type of carotenoid. It is also possible to use as an enhancer to treat cases of low intake or hypovitaminosis of vitamin A. It is known that some carotenoids, such as β-carotene, have an orange color and are naturally present in some foods such as carrots and pumpkins. It is a provitamin A agent, acting as a precursor of retinol; in this way, it can be produced by microorganisms and work as an additive in other foods that naturally do not have retinol or provitamin A, such as juices, butter, creams, milk, and even sweets [3].

## 8. Conclusions

Microbial pigments are derived from the most different species in nature, including microbes already adapted to human interaction without presenting pathogenicity. They can even play roles as colorants and additives to improve formulations with their bioactive properties (antioxidant, antimicrobial, anti-inflammatory, among others). More investments in the segment, regulations, and legislation are necessary for their production, application, and commercialization.

Their use until the present time is underestimated, and they present a range of alternatives to replace traditional mineral and synthetic dyes. Even though there are still some barriers to its large-scale production, its application to the most diverse industrial sectors, such as the pharmaceutical, food, textile, and cosmetics industries, is covered with possibilities. The advancement of integrated molecular studies, facilitated by the confluence of bioinformatic methodologies and omics platforms, is poised to address the bottleneck challenges associated with pigment production. Utilizing agro-waste and industrial byproducts facilitates and enhances the bioprocess and contributes to cost reduction.

Biotechnology can be the key to sustainable development, thus anticipating and solving consumption-related problems. The process can continually be optimized to reduce downstream processes and environmental impacts, produce more stable pigments, use by-products from other sectors as fermentation substrates, and discover new pigments and their applications. There are still steps to be overcome, but microbial pigments offer economic, environmental, industrial, and significant commercial appeal potential, and with investment they can become the primary source of natural pigments and help reduce the use of synthetic pigments in the industry.

## Figures and Tables

**Figure 1 microorganisms-11-02920-f001:**
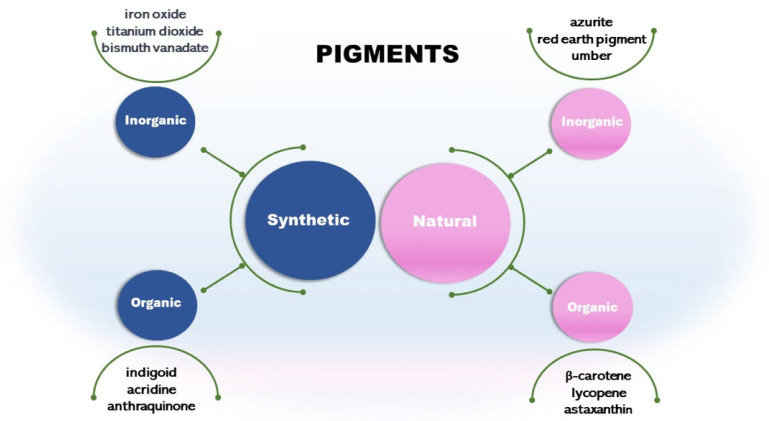
Major types of synthetic and natural pigments.

**Figure 2 microorganisms-11-02920-f002:**
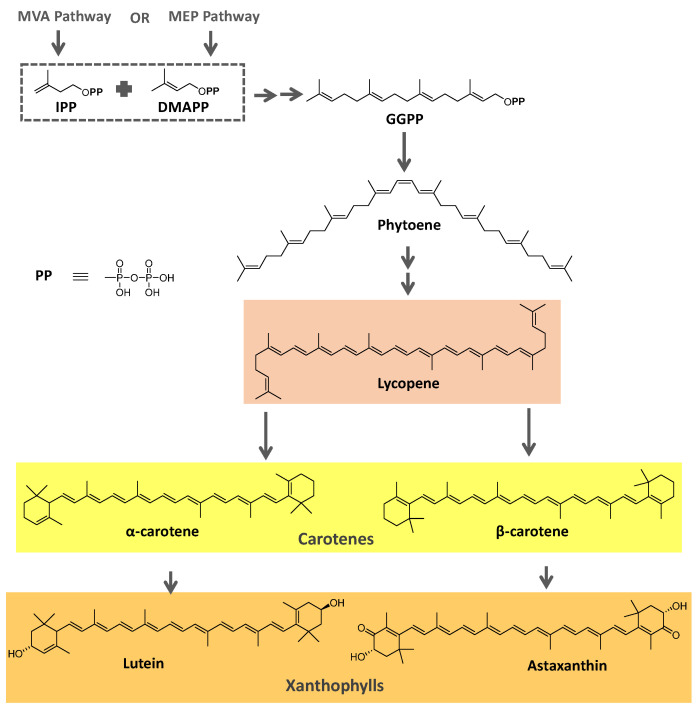
Schematic diagram of carotenoids biosynthetic pathways. DMAPP = dimethylallyl diphosphate; GGPP = geranylpyrophosphate; IPP = isopentenyl diphosphate; MVA = mevalonic acid; MEP = 2-*C*-methyl-D-erythritol-4-phosphate.

**Figure 3 microorganisms-11-02920-f003:**
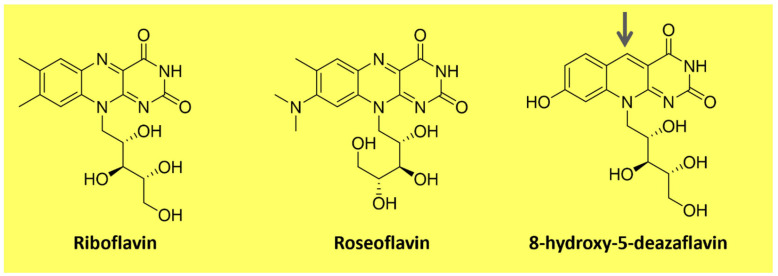
Examples of flavins. The arrow points to the carbon atom at position five in the 5-deazaflavin which replaces a nitrogen atom.

**Figure 4 microorganisms-11-02920-f004:**
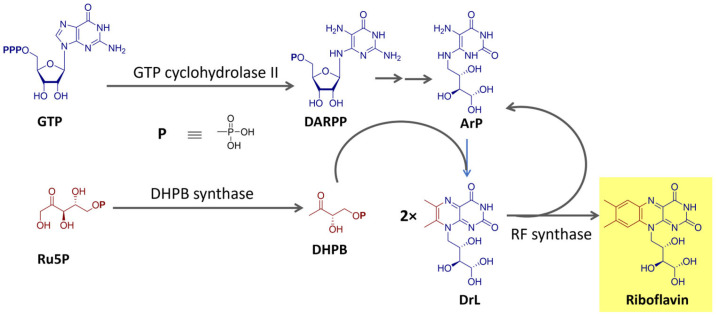
Schematic diagram of riboflavin biosynthesis pathway. ArP = 5-amino-6-ribitylamino-2,4(1H,3H)-pyrimidinedione; DARPP = 2,5-diamino-6-ribosylamino-4(3H)-pyrimidinone 50-phosphate; DHBP = 3,4-dihydroxybutanone 4-phosphate; DrL = 6,7-dimethyl-8-ribityllumazine; GTP = purine guanosine triphosphate; RF = riboflavin; Ru5P = ribulose-5-phosphate.

**Figure 5 microorganisms-11-02920-f005:**
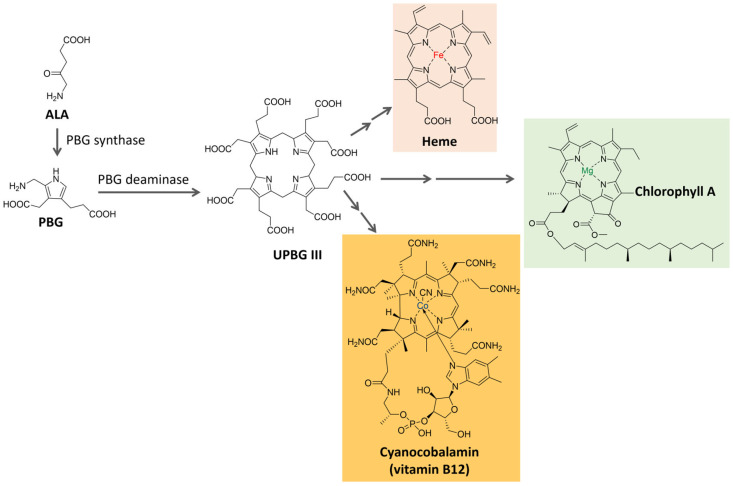
Simplified schematic diagram of the biosynthetic pathway of tetrapyrrole macrocyclic compounds. ALA = δ-aminolevulinic acid; PBG = porphobilinogen; UPBG III = uroporphyrinogen III.

**Figure 6 microorganisms-11-02920-f006:**
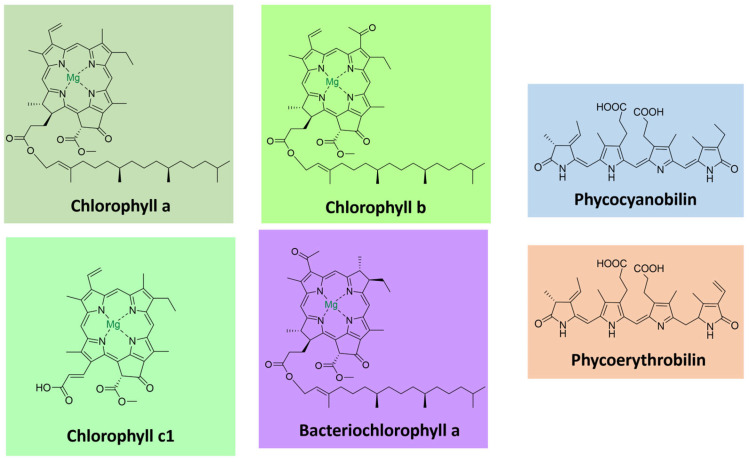
Structure of chlorophylls a, b, and c1, bacteriochlorophyll, and phycobilins (phycocyanobilin and phycoerythrobilin).

**Figure 7 microorganisms-11-02920-f007:**
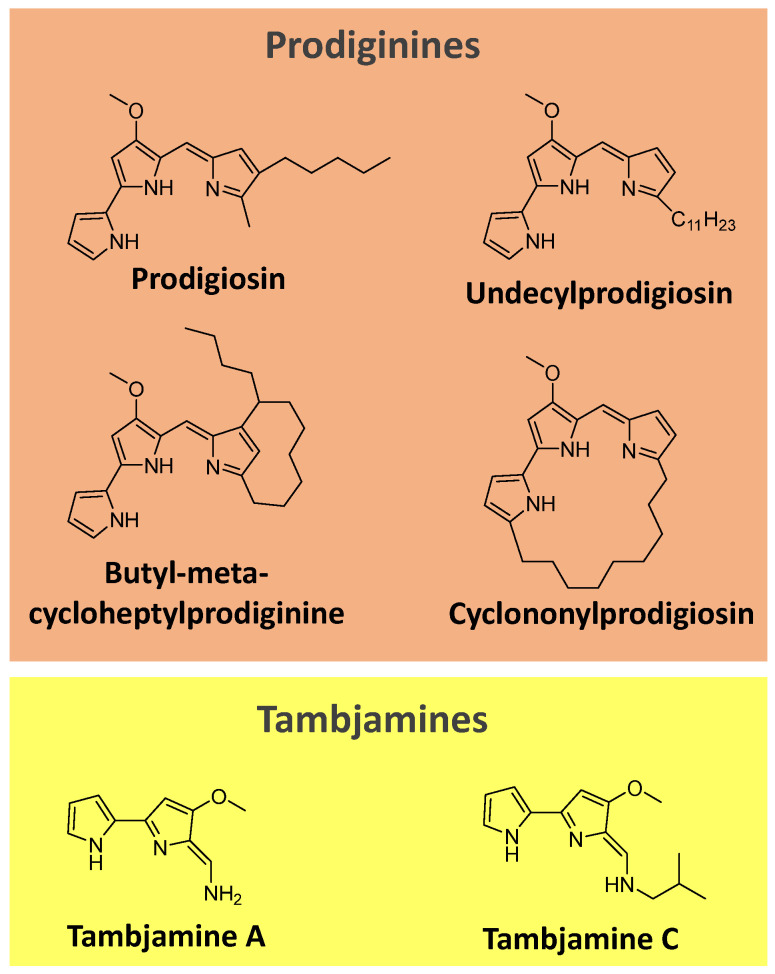
Structures of prodiginines and tambjamines.

**Figure 8 microorganisms-11-02920-f008:**
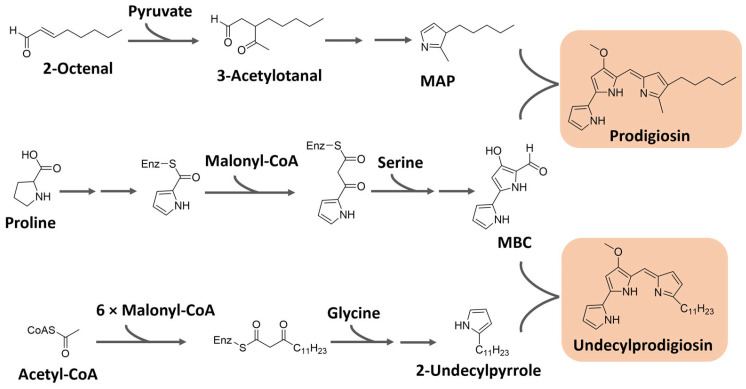
Simplified schematic diagram of biosynthetic pathways of prodiginines. Enz = cluster of enzymes. [90]. MAP = 2-methyl-3-n-amyl-pyrrole; MBC = 4-methoxy-2-2′-bipyrrole-5-carbaldehyde.

**Figure 9 microorganisms-11-02920-f009:**
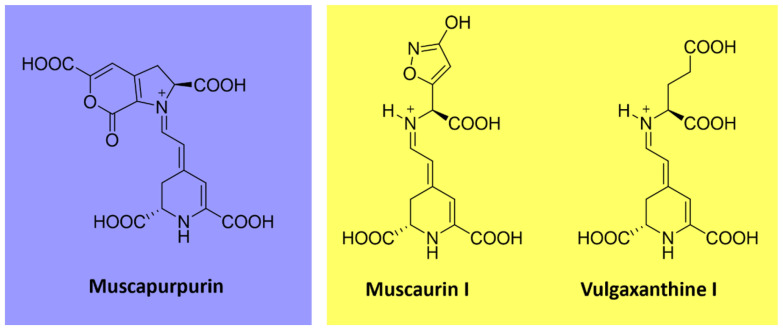
Chemical structure of the pigments muscapurpurin, muscaurin I, and vulgaxanthine I from *Amanita muscaria*.

**Figure 10 microorganisms-11-02920-f010:**
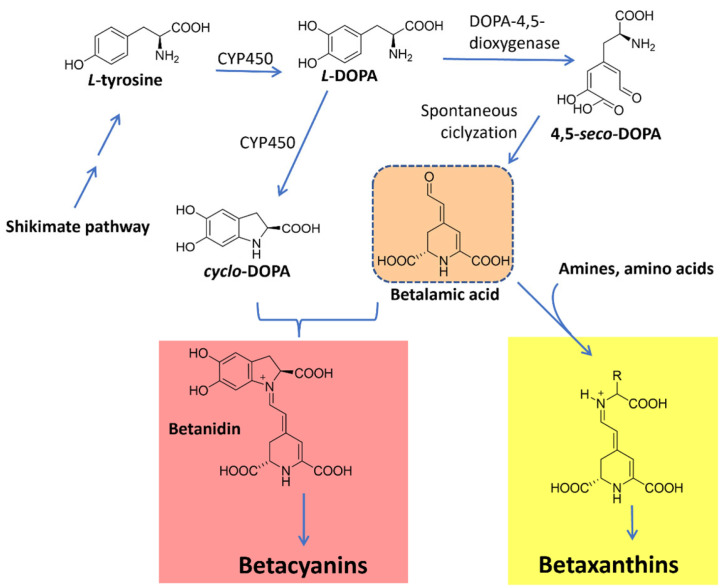
Schematic diagram of betalain biosynthetic pathways based on [101,105]. CYP450 = cytochrome P450.

**Figure 11 microorganisms-11-02920-f011:**
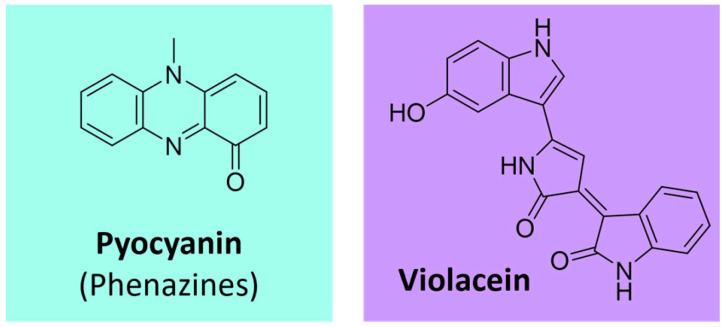
Structures of pyocyanin and violacein.

**Figure 12 microorganisms-11-02920-f012:**
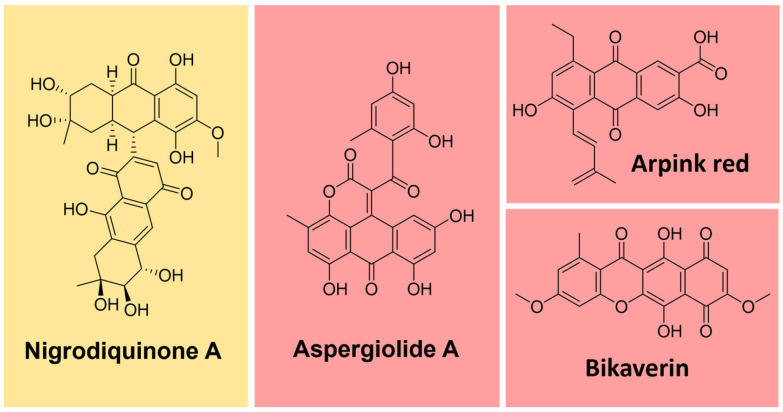
Structure of some quinone pigments.

**Figure 13 microorganisms-11-02920-f013:**
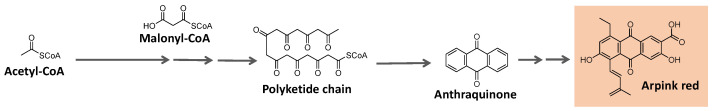
Simplified schematic diagram of biosynthetic pathways of anthraquinones from the acetyl-CoA/malonyl-CoA pathway.

**Figure 14 microorganisms-11-02920-f014:**
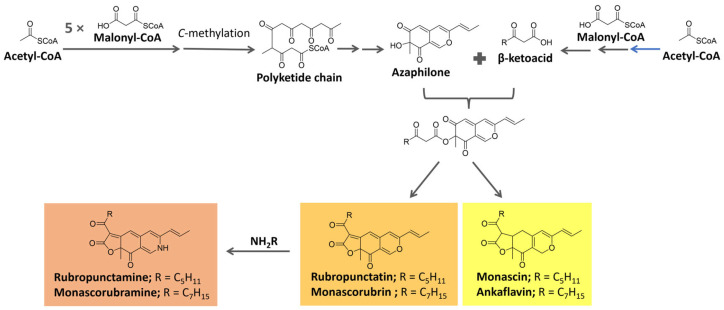
Simplified schematic diagram of biosynthetic pathways of azaphilone pigments. (Based on [119,122]).

**Figure 15 microorganisms-11-02920-f015:**
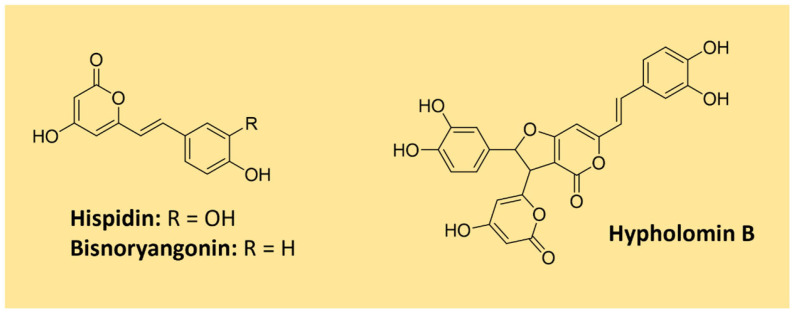
Styrylpyrone pigments from fungi.

**Figure 16 microorganisms-11-02920-f016:**
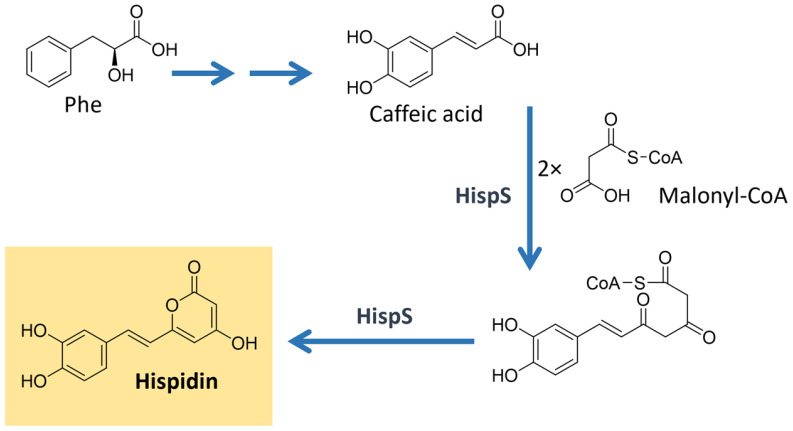
Proposed biosynthetic pathway for hispidin. Phe = phenylamine; HispS = hispidin synthase [123,125].

**Figure 17 microorganisms-11-02920-f017:**
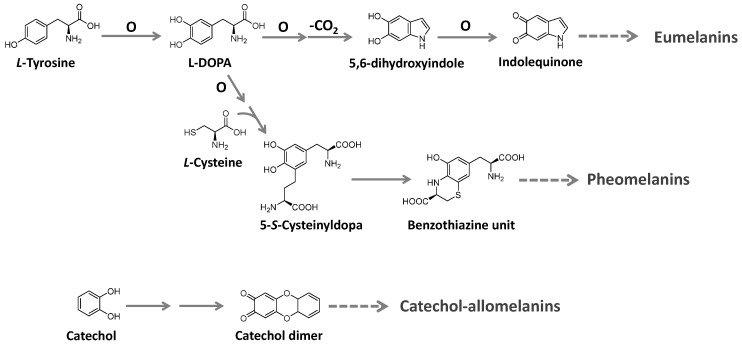
Simplified schematic diagram of biosynthetic pathways of melanins based on [129].

**Table 1 microorganisms-11-02920-t001:** Advantages and disadvantages of microbial, vegetable, and mineral pigments [37,38,39].

Parameters	Mineral	Vegetable	Microbial
Price/production costs	✓	X	X
Biodegradability	X	✓	✓
Contamination free	X	X	✓
Renewable	X	✓	✓
Use of substrate	X	✓	✓
Resistant to climate change	✓	X	✓
Genetic manipulation	X	✓	✓

✓—positive; X—negative.

**Table 2 microorganisms-11-02920-t002:** Carotenoids from microbial sources and their structural formula (adapted from [52,63,64,65].

Microorganisms	Carotenoids and Their Structural Formula
Bacteria: *Rhodococcus maris* *Micrococcus roseus* *Microbacterium* sp. *LEMMJ01* *Gordonia jacobaea* *Bradyrhizobium* sp. Cyanobacteria: *Anabaena variabilis* *Aphanizomenon flos-aquae* *Nostoc commune*	Canthaxanthin 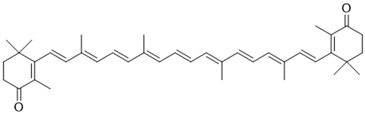
Bacteria: *Microbacterium* sp. *LEMMJ01* *Paracoccus* sp. *Halobacterium salinarium* Microalgae: *Haematococcus pluvialis* Yeast: *Phaffia rhodozyma* (*Xanthophyllomyces dendrohous*)	Astaxanthin 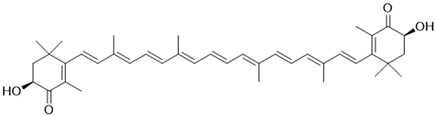
Bacteria: *Pseudomonas putida* *Mycobacterium kansasii* Microalgae: *Dunaliella salina* *Spirulina* Filamentous fungi: *Blakeslea trispora* *Phycomyces blaskeleeanus* *Mucor circinelloides* Yeast: *Rhodotorula glutinis*	β-carotene 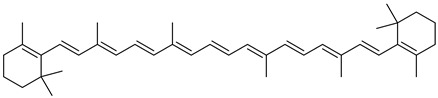

**Table 3 microorganisms-11-02920-t003:** Microbial pigments: colors and applications.

Pigment/ Chemical Class	Color	Biological Activity	Microorganism	Status	References
^99m^Tc-Anthraquinone pigment complex	Red	Antioxidant and anticancer	*Talaromyces purpureogenus* (Fungi)	Research pharmaceutical use	[136]
Indigoidine	Blue	Antioxidant	*Corynebacterium insidiosum*; *Corynebacterium glutamicum*; *Streptomyces chromofuscus* (Bacteria)	Research and industrial production	[25,46,137]
Melanin	Black	Antimicrobial, antibiofilm, and antioxidant	*Colletotrichum lagenarium*, *Aspergillus fumigatus*, *Aureobasidium melanogenum* (fungi) *Vibrio cholerae*, *Shewanella colwelliana*, *Alteromonas nigrifaciens*, (Bacteria)	Research	[138,139,140]
Monascorrubramin	Red	Antioxidant and anticancer	*Monascus* sp. (Fungi)	Research and industrial production	[141,142]
Prodigiosin	Red	Anticancer, antimicrobial, and immunosuppressant	*Serratia marcescens*; *Pseudoalteromonas rubra* (Bacteria)	Research and industrial production	[3,91,143]
Chlorophylls	Green	Improving immune system antioxidant and anticancer	*Chlorella* sp. *Scenedesmus dimorphus* *Chlamydomonas reinhardtii* (microalgae)	Research and industrial production	[144]
Astaxanthin	Red	Antioxidant, photoprotector, anti-inflammatory, anticancer, antimicrobial, and antihyperlipidemia that increases serum adiponectin.	*Haematococcus pluvialis* *Chlorella* sp. (microalgae)	Research and industrial production	[145,146,147]
Pyocanin	Blue-green	Antioxidant	*Pseudomonas aeruginosa* (Bacteria)	Research	[148]
Riboflavin	Yellow-orange	Nutritional supplement	*Bacillus* sp.; *Ashbya gossypii* (Bacteria)	Research and industrial production	[149,150]

**Table 4 microorganisms-11-02920-t004:** Recent findings in the literature of alternative agro-waste and byproducts for microbial pigment production.

Alternative Agro-Waste/ByProducts	Pigment Studied	Microorganisms	Pigment Yield	References
Wheat wastes	Astaxanthin	*Yamadazyma guilliermondii*, *Xanthophyllomyces dendrorhous*, *Yarrowia lipolytica* and *Sporidiobolus salmonicolor*	109.23 μg/gram of waste maximum	[178]
Glycerol, Corn steep liquor, parboiled rice water	Carotenoids (B-carotene)	*Sporidiobolus pararoseus*	843 μg/L total (346 ug/L b-carotene)	[179]
Rice straw hydrolysate with glucose medium	Extracellular azaphilones	*Monascus* sp. (mutant strain)	20.86 U/mL	[180]
Food industry wastewater	Carotenoid	*R. mucilaginosa*	810 μg/g	[181]
Pineapple peel waste	-	*Talaromyces albobiverticillius*	0.523 mg/g	[182]
Corncob Hydrolysates	monascorubrin and rubropunctamine	*T. atroroseus*	16.17 OD_500 nm_	[183]
Orange and grape wastes	β-carotene	*R. glutinis*	5.9 g/L maximmum	[184]

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
