# Peer review of "Microbial Pigments: Major Groups and Industrial Applications"

_microorganisms, 2023, doi:10.3390/microorganisms11122920_

Round 1

Reviewer 1 Report

Comments and Suggestions for Authors

This review describes the structure of microbial pigments, the advantages of their use, and their applications in different fields, which will help further research on microbial pigments. The logic of the manuscript is confusing, the data collection is not rigorous enough, and the current content of the manuscript is insufficient to support the title.

Only some of the bugs are listed below, and the authors are kindly requested to reread the whole manuscript carefully and check it in detail.

Major comments

Phenol compounds from specific fungal species are also a well-known class of pigments, please refer to and cite the following literature:

A: Styrylpyrone-class compounds from medicinal fungi Phellinus and Inonotus spp. and their medicinal importance. J. Antibiot. 2011, 64, 349 J. Antibiot. 2011, 64, 349-359.

B: Diverse Metabolites and Pharmacological Effects from the Basidiomycetes Inonotus hispidus, Antibiotics 2022, 11(8), 1097

The gene clusters for the biosynthetic processes shown in Figures 6, 8, 9, and 11 should also be shown;

Minor errors

The doi numbers of the references are unnecessary

Figure 1. - Major types of synthetic and natural pigments.

Line 344  [102,103.104].

Comments on the Quality of English Language

It is OK.

Author Response

this review describes the structure of microbial pigments, the advantages of their use, and their applications in different fields, which will help further research on microbial pigments. The logic of the manuscript is confusing, the data collection is not rigorous enough, and the current content of the manuscript is insufficient to support the title.

Only some of the bugs are listed below, and the authors are kindly requested to reread the whole manuscript carefully and check it in detail.

Major comments 

1.Phenol compounds from specific fungal species are also a well-known class of pigments, please refer to and cite the following literature:

A: Styrylpyrone-class compounds from medicinal fungi Phellinus and Inonotus spp. and their medicinal importance. J. Antibiot. 2011, 64, 349 J. Antibiot. 2011, 64, 349-359.

B: Diverse Metabolites and Pharmacological Effects from the Basidiomycetes Inonotus hispidus, Antibiotics 2022, 11(8), 1097 

 As suggested by the reviewers, the references were added.

2.The gene clusters for the biosynthetic processes shown in Figures 6, 8, 9, and 11 should also be shown.

Figure 6- In this figure, there are only known chlorophyll structures, not biosynthesis pathways.

 Figure 8 -   prodigiosin is a Prodiginines, and the  clusters were added in the revised text, in lines 381-384

Figure 9 melanin  a nao figure 17 was based on the reference Solano et al. 2014, and the clusters were added in the revised text, in lines 543-547;

Solano, F. Melanins: Skin Pigments and Much More—Types, Structural Models, Biological Functions, and Formation Routes. 939  New J. Sci. 2014, 2014, 1–28, doi:10.1155/2014/498276. 

figure 11, now figure 14, is a biosynthesis of  azaphilone

the  clusters were added in the revised text, in lines 489-492  

  was based on two references about the pathwayLiu, L.; Zhao, J.; Huang, Y.; Xin, Q.; Wang, Z. Diversifying of Chemical Structure of Native Monascus Pigments. Front. Microbiol. 957    2018, 9, 1-13.doi:10.3389/fmicb.2018.03143.

Chen, W.; Chen, R.; Liu, Q.; He, Y.; He, K.; Ding, X.; Kang, L.; Guo, X.; Xie, N.; Zhou, Y.; et al. Orange, Red, Yellow: Biosynthesis  960   of Azaphilone Pigments in Monascus Fungi. Chem. Sci. 2017, 8, 4917–4925 .doi:10.1039/C7SC00475C.

 Minor errors

  1. The doi numbers of the references are unnecessary.
  2.  The doi of references was removed.

4.Figure 1. - Major types of synthetic and natural pigments.

    The error of the figure was not pointed out, and for the authors, the figure is correct.

Reviewer 2 Report

Comments and Suggestions for Authors

I have read the manuscript, it is correct, but it does not contain anything new or revealing. This topic has been discussed in many review articles published in last years, including:

Microbial pigments: learning from the Himalayan perspective to industrial applications. Doi: https://doi.org/10.1093/jimb/kuac017

Microbial Pigments: Natural Colorants and their Industrial Applications. Doi: https://doi.org/10.20546/ijcmas.2021.1005.071

Natural Pigments of Microbial Origin. Doi: https://doi.org/10.3389/fsufs.2020.590439

Biopigments of Microbial Origin and Their Application in the Cosmetic Industry. Doi: https://doi.org/10.3390/cosmetics10020047

Microbial pigments as an alternative to synthetic dyes and food additives: a brief review of recent studies. Doi: https://doi.org/10.1007/s00449-021-02641-4

Microbial Pigments in the Food Industry—Challenges and the Way Forward. Doi: https://doi.org/10.3389/fnut.2019.00007

Research progress, challenges, and perspectives in microbial pigment production for industrial applications - A review. Doi: https://doi.org/10.1016/j.dyepig.2022.110989

Multifaceted Applications of Microbial Pigments: Current Knowledge, Challenges and Future Directions for Public Health Implications. Doi: https://doi.org/10.3390/microorganisms7070186

and many others. What is new in the article prepared by the authors? The review article should clearly indicate its innovation and future perspectives of the topic.

Reviewer 3 Report

Comments and Suggestions for Authors

This review meticulously examines a wide array of pigments derived from microorganisms, emphasizing their applicability in biotechnological realms. A critical synthesis of existing work, encompassing analyses of over 180 literary sources, has been conducted to consolidate findings pertaining to pigment properties. While certain segments of the review, such as Section 6, are commendably executed, there are areas identified for potential enhancement and further clarification.

Introduction  (Section 2-5 Revision Suggestion):

The introduction, currently extensive, could be made more concise and focused, ensuring it offers a clear and logical opening to the subject matter.

Scalability and Technological Viability:

Clarification and in-depth discussion are required concerning the scalability of technologies involved in pigment production from microorganisms. Critical analysis should assess whether the current technological advancements align with contemporary market demands and expectations.

Influential Factors in Pigment Formation:

A comprehensive exploration of factors influencing pigment formation, such as light and temperature, is imperative. The review should offer a detailed examination of how these variables may impact pigment characteristics and yield

Future Prospects and Enhancement Areas:

A focused discussion on future trajectories and possible enhancement areas in the exploitation of microbial pigments is essential. This includes potential strategies for optimizing pigment production and utilization in various biotechnological applications.

Potential Applications in Animal Husbandry:

The review should also contemplate and elaborate on the prospective utilization of microbial pigments in animal husbandry, exploring the feasibility and benefits of such applications.

A coherent conclusion that encapsulates the key findings, insights, and recommendations gleaned from the review is necessary to provide readers with a synthesized understanding of the field’s current state and future directions.

Author Response

Comments and Suggestions for Authors

This review meticulously examines a wide array of pigments derived from microorganisms, emphasizing their applicability in biotechnological realms. A critical synthesis of existing work, encompassing analyses of over 180 literary sources, has been conducted to consolidate findings pertaining to pigment properties. While certain segments of the review, such as Section 6, are commendably executed, there are areas identified for potential enhancement and further clarification. 

Introduction  (Section 2-5 Revision Suggestion):

The introduction, currently extensive, could be made more concise and focused, ensuring it offers a clear and logical opening to the subject matter.

 As suggested by the reviewer, the introduction was  modified  and reduced but  still contains relevant information.  

Scalability and Technological Viability

Clarification and in-depth discussion are required concerning the scalability of technologies involved in pigment production from microorganisms. Critical analysis should assess whether the current technological advancements align with contemporary market demands and expectations.

 New text containing all the requested information was added to the review, and new sections were created.

Influential Factors in Pigment Formation: A comprehensive exploration of factors influencing pigment formation, such as light and temperature, is imperative. The review should offer a detailed examination of how these variables may impact pigment characteristics and yield.

The review was augmented to 46 pages  ( from 26)  to include all the reviewers' suggestions. We did not create a new section for this specific theme. However, in item 8,3- pigment extraction, we added information about these factors. 

Future Prospects and Enhancement Areas: A focused discussion on future trajectories and possible enhancement areas in the exploitation of microbial pigments is essential. This includes potential strategies for optimizing pigment production and utilization in various biotechnological applications.

 New sections were added to answer these questions as requested by the reviewer. 

Potential Applications in Animal Husbandry:The review should also contemplate and elaborate on the prospective utilization of microbial pigments in animal husbandry, exploring the feasibility and benefits of such applications.

More information was added about this in pigment in the food industry. 

A coherent conclusion that encapsulates the key findings, insights, and recommendations gleaned from the review is necessary to provide readers with a synthesized understanding of the field’s current state and future directions.

 The conclusion was modified in order to attend to the reviewer.

Round 2

Reviewer 1 Report

Comments and Suggestions for Authors

The careful revision of the manuscript by the author has significantly improved its quality. As a review, the current version is more comprehensive. However, the redundant DOI numbers have not been removed, but this does not prevent the manuscript from being considered for publication.

Comments on the Quality of English Language

It is OK.

Author Response

The careful revision of the manuscript by the author has significantly improved its quality. As areview, the current version is more comprehensive. However, the redundant DOI numbers have ot been removed, but this does not prevent the manuscript from being considered forpublication.

 Thank you for the comments that improved the review. The DOi number was removed in this last revision.

Reviewer 2 Report

Comments and Suggestions for Authors

The authors did not clearly respond to the reviewer's comments, they only provided me with general information. I still don't see anything new in this literature review.

Reviewer 3 Report

Comments and Suggestions for Authors

Despite the fact that the authors did not respond to all of the reviewer’s comments, in my opinion the corrections made are satisfactory. I have no serious comments. The review has become more interesting and, in my opinion, will find its readers.

Author Response

no issues were raised by this reviewer